# Influence of γ-Radiation on Mechanical Stability to Cyclic Loads Tubular Elastic Matrix of the Aorta

**DOI:** 10.3390/jfb13040192

**Published:** 2022-10-17

**Authors:** Alexander Yu. Gorodkov, Yuriy M. Tsygankov, Alexey D. Shepelev, Sergey V. Krasheninnikov, Shota T. Zhorzholiani, Andrey V. Agafonov, Vissarion G. Mamagulashvili, Dmitriy V. Savinov, Timur Kh. Tenchurin, Sergey N. Chvalun

**Affiliations:** 1A.N. Bakulev National Medical Research Center for Cardiovascular Surgery, Rublevskoye Highway 135, 121552 Moscow, Russia; 2National Research Centre “Kurchatov Institute”, Akademika Kurchatova pl. 1, 123182 Moscow, Russia

**Keywords:** vascular prostheses, copolymers of vinylidene fluoride with hexafluoropropylene, electrospinning, mimic the mechanical properties of native vessels, restoration of the neovessel

## Abstract

**Highlights:**

Radiation treatment of tubular prostheses based on a mixture of copolymers of VDF/HFP achieved mechanical behavior of the aorta. Optimum mechanical properties of tubular prostheses were achieved at a dose of 0.3 MGy. The mechanical properties of the fabricated vessel created favorable conditions for restoration of the neovessel.

**Abstract:**

A significant drawback of the rigid synthetic vascular prostheses used in the clinic is the mechanical mismatch between the implant and the prosthetic vessel. When placing prostheses with radial elasticity, in which this deficiency is compensated, the integration of the graft occurs more favorably, so that signs of cell differentiation appear in the prosthesis capsule, which contributes to the restoration of vascular tone and the possibility of vasomotor reactions. Aortic prostheses fabricated by electrospinning from a blend of copolymers of vinylidene fluoride with hexafluoropropylene (VDF/HFP) had a biomechanical behavior comparable to the native aorta. In the present study, to ensure mechanical stability in the conditions of a living organism, the fabricated blood vessel prostheses (BVP) were cross-linked with γ-radiation. An optimal absorbed dose of 0.3 MGy was determined. The obtained samples were implanted into the infrarenal aorta of laboratory animals—Landrace pigs. Histological studies have shown that the connective capsule that forms around the prosthesis has signs of high tissue organization. This is evidenced by the cells of the fibroblast series located in layers oriented along and across the prosthesis, similar to the orientation of cells in a biological arterial vessel.

## 1. Introduction

According to the World Health Organization, cardiovascular disease ranked first among the diseases leading to death. Most diseases of the aorta, coronary, and peripheral arteries require surgical intervention using prosthetic blood vessels. Currently, synthetic rigid vascular prostheses made of polytetrafluoroethylene (PTFE), Dacron^TM^, or polyurethane are widely used in cardiovascular surgery [1,2]. The main complications of the use of rigid vascular prostheses are neointimal hyperplasia in the area of the distal anastomosis and post-stenotic dilatation of the vessel in the distal segment along the direction of blood flow [3]. The “gold standard” for bypass surgery on coronary and peripheral arteries of small diameter are autologous blood vessels [4,5,6]. However, during reconstructive operations on peripheral arteries, especially on the aorta, autologous blood vessels of the required diameter are not always available. The use of decellularized allografts is limited due to complications associated with their calcification, thrombosis, and aneurysm. Therefore, at present, on the basis of natural and synthetic polymers, tubular scaffolds are manufactured, seeded with cells that act as an analogue of the natural extracellular matrix, and also control the process of tissue development [7,8,9]. However, despite the achievements in the field of tissue engineering, no graft has yet been adopted in routine clinical practice [4,6]. One of the reasons is that many of the developed structures do not have the necessary mechanical properties.

In previously published works, it was proven that the blood flow in the heart and main arteries has a spiral nature, which is described by exact solutions of non-stationary hydrodynamic equations for the class of tornado-like viscous fluid flows [10]. The non-stationary regime of blood flow requires that, in order to maintain the necessary geometric configuration of the flow channel, its boundaries should be sufficiently flexible, which is possible only for radially elastic walls. Under these conditions, the swirling flow retains its structure without the formation of congested and separation zones, and the continuum of the boundary layer free from shear stresses is also preserved. In our previous studies [11], it was shown that prototypes of aortic prostheses obtained by electrospinning of a blend of various VDF/HFP copolymers in 0.85/0.15 proportion were closest ones in biomechanical behavior to the native aorta. The copolymers varied in the content of VDF and HFP units—VDF/HFP (8:1) and VDF/HFP (3:1). However, a serious problem encountered lies that lies in the low mechanical stability of these implants under living conditions. The implanted grafts increased in diameter over time. The process began a few hours after installation. The main goal of recent study is in eliminating this drawback.

It was obvious that the insufficient biomechanical stability of the prepared prototypes of blood vessel prostheses is associated with insufficient strength of the interfiber bonds which increased by chemical crosslinking. The process of crosslinking such materials was first described in [12]. It was shown that the γ-radiation of vinylidene fluoride copolymers is accompanied by both crosslinking and degradation, which is explained by the presence of both fully fluorinated units and methylene groups in the copolymer structure [13]. It was found [12] that the destruction process prevails at irradiation of fibrous materials in air, while in an inert atmosphere the crosslinking dominates. Thus the proposed manufacturing of a tubular elastic aortic matrix persistent to cyclic loads, should consist at least two stages: the manufacture of the matrix by electrospinning from a blend of copolymers VDF/HFP (8:1) and VDF/HFP (3:1), as was described in details in [14,15], and the γ-radiation of polymer graft, providing simultaneously the implant sterilization. It was shown that at an absorbed dose less than 0.3 MGy, the polymer crosslinking to a large extent takes place, without a substantial loss of its strength [12]. However the detailed research on mechanical behavior of artificial graft prepared by electrospinning irradiated by different doses, its comparison the with performance of native aorta, the stability of prostheses, both in testing machine and during implantation into a living organism is needed for the development of advanced aorta prostheses and their implementation in clinical practice.

## 2. Materials and Methods

### 2.1. Electrospinning of Blood Vessel Prostheses

The VDF/HFP (8:1) and VDF/HFP (3:1) copolymers were purified by the reprecipitation method, with acetone + methyl ethyl ketone 1:1 mixture as a solvent, and chloroform as a precipitant; the purity of the solvents was “analytical grade”. Residual solvent content was controlled by the thermogravimetric analysis. Polymer solutions in acetone and methyl ethyl ketone for electrospinning were prepared. To increase the electrical conductivity of spinning solutions 10 w%. of ethyl alcohol was added. 

Aortic prosthesis matrices were manufactured using the Professional Electrospinning unit V 2.0 (Doxa Microfluidics, Málaga, Spain). The deposition was carried out on a receiving conical metal electrode with a diameter of 9.4–10.4 mm, a length of 100 mm, and a surface roughness of 7. The electrode rotated at a speed of 64 rpm. A nozzle had a “bell” design [14], with a resistance of 10 mm of water. The nozzle performed a reciprocating motion at a speed of 1 cycle per 1 min. A high negative voltage was applied to the polymer solution. Samples were prepared at a temperature of 23–25 °C and a humidity of 40–55%. The solutions of two copolymers VDF/HFP (8:1) and VDF/HFP (3:1) in ) with a concentration of 8.9% were electrospun at the voltage of 19 kV, a working distance of 23 cm, and a solution flow of 20 mL/h.

The resulting samples were dried to a constant weight at a vacuum of 10^−3^ Torr. Radiation treatment of the fabricated matrices was carried out at a temperature of 23 °C in glass ampoules evacuated to a vacuum of 10^−3^ Torr. Irradiation was carried out using a source of γ-radiation Co60. Dose rate of 4.9 Gy/s. Absorbed doses were 0.1, 0.2, 0.3, and 0.4 MGy.

### 2.2. Mechanical Properties of BVP

Study the mechanical properties and stability of the prepared BVP samples in the radial direction was performed on “ring” type samples of 10 mm width both in air (“dry” samples) and in a phosphate buffer (“wet” samples). The samples were freely fixed on cylindrical supports 2 mm in diameter. Due to instrumental limitations, the deformation rate was significantly lower than the physiological one (200%/min). Before testing, “dry” matrices were conditioned at 23 ± 2 °C for 24 h and “wet” matrices were kept in phosphate buffer at 23 ± 2 °C for 24 h. Specific mechanical characteristics were calculated using the measured linear dimensions. The sample thickness was determined as twice the BVP wall thickness. The initial length of the sample was determined as the distance between the centers of the supports.

As a result of the tests, tensile curves were obtained in the load– elongation coordinates. To determine the curve part corresponding to the physiological region (40–320 mm Hg) on the deformation curves, they were brought to the form of pressure–deformation. The current load was converted into pressure using the Laplace equation, as was done in [16]. On the measured dependences, the upper and lower limits of loads corresponding to the pressure range of 80–160 mm Hg were determined. The limits were used in cyclic tests later.

Cyclic studies (characterizing the mechanical stability of the investigated BVP) were carried with a constant speed of movement of the clamp. The number of load–unload cycles for each BVP sample was 10,000. During testing in the physiological range of loads corresponding to 80–160 mm Hg, the change in maximal strains was tracked. Thus, the final result of the cyclic tests was a level of “steady” deformations in the range of physiological pressures, and in accordance with ISO 5840:2005 ISO 7198:1998, the compliance of prototype aortic prostheses was calculated.

The value of compliance (C_0_) was calculated on the basis of pulse changes in the radial size in the middle part of the prosthesis according to the equation:$$C_{0}=\frac{(R_{p2}-R_{p1})/R_{p1}}{p_{2}-p_{1}}\times 10^{4}$$
where *R* is the radius of the prosthesis; *p*_1_ is the diastolic pressure value; and *p*_2_ is the systolic pressure value (mmHg). Compliance was expressed as a percentage change in diameter per 1 mm Hg. 

### 2.3. In Vivo Study of Vascular Prostheses Samples

All procedures with animals were carried out in strict adherence to the requirements of the Local Ethics Committee on Biomedical Research (NRC «Kurchatov Institute»; Protocol #09/KПБ-21, 28 December 2021). Samples were implanted into the infrarenal part of the aorta in three pigs (females of the Landras breed weighing 45 kg) for a period of 30 days. Operations were carried out under general anesthesia, access to the infrarenal part of the aorta was carried out postero-peritonially from the left side. The pulse changes in the radial prosthesis size were measured by contrast angiography (mobile angiographic facility of Ziehm Vision R, Germany, the contrast—Omnipaque 300 mg/mL). Angiography was performed on the day of the operation, and on the 3rd and 30th day after implantation. The pulse pressure in the aorta of all pigs ranged from 65 to 140 mm Hg.

After euthanasia of the animals, the vascular prostheses samples were extracted, and macroscopic, histological, and immunohistochemical studies of the capsule were carried out.

All experiments were carried out in accordance with the requirements of “OECD principles of good laboratory practice” (OECD Guide 1: 1998, IDT).

## 3. Results

### 3.1. Characterization of the Grafts

A tubular prostheses with a length of 4 cm, a diameter of 10 mm and a wall thickness of 0.8 mm was obtained by electrospinning (Figure 1a). In Figure 1b, the scanning electron micrograph from the surface vascular graft showed fibers with a diameter of 1.7 ± 0.5 μm. Porosity (85%) providing the morphology suitable for cell proliferation, blood cell penetration and necessary level of mechanical characteristics of implant.

### 3.2. Mechanical tests

The mechanical properties of the original and irradiated BVP prototypes prepared from a 0.85/0.15 blend of VDF/HFP (3:1) and VDF/HFP (8:1) copolymers were studied. The mechanical properties of the original and irradiated samples (“dry” and “wet”) are shown in Table 1.

From the test results of the “dry” samples, it can be seen that at a dose of 0.4 MGr, the mechanical characteristics changed dramatically. The strength dropped by two times, the break deformation – by four times, while the tensile modulus increased by almost 3 times. At the same time, the mechanical characteristics of samples irradiated by 0.1–0.2 MGy are very close to initial ones indicating to insufficient crosslinking of polymer matrix.

In Figure 2, the initial sections of the deformation curves corresponding to the studied physiological range are shown.

The data obtained are in good agreement with the previously published results [12]. It can be seen that in the initial BVP samples, when moistened some decrease in strength and elastic modulus occurred. The constancy of the characteristics of irradiated grafts indicates the appearance of a chemical cross-link between the fibers of the matrices. A comparison of the properties of a material with an absorbed dose of 0.1 MGy and a material with a dose of 0.3 MGy showed that active degradation processes were already taking place in the latter, but they are not critical yet. At the same time, it can be seen from the figures that the elasticity of the sample with a dose of 0.3 MGy in the “wet” state is quite close to the elasticity of the original sample.

From the measured deformation curves, the physiological load limits for cyclic tests were determined. In these ranges, the original and modified samples were studied in the “dry” and “wet” states. The test results are shown in Figure 3.

Analyzing the effect of the absorbed dose value, it can be noted that in the initial samples, when moistened, the average change in the BVP diameter increased both at 80 and 160 mm Hg. For irradiated samples, the dose effect on the mechanical behavior was observed. At 0.1 MGy, the relative change in the diameter of the “wet” sample is considerably lower than that of the dry one (under the same test conditions). At 0.3 MGy, the picture is qualitatively the same. The quantitative characteristic used in assessing the mechanical properties of BVP is compliance. Data on changes in the compliance of “dry” and “wet” BVP, depending on the absorbed dose, are given in Figure 4.

It can be seen that the compliance of the BVP decreased with an increase in the absorbed dose (in the studied range). The negative change in compliance in the “dry” samples with increasing dose was noticeably greater than in the “wet” ones. In a previous study [14], a comparison was made of the elastic moduli of BVPs made by electrospinning from various polymers and their blends. It was shown that the elasticity of the BVP sample made from a blend of VDF/HFP (3:1) and VDF/HFP (8:1) was closest to the elasticity of the native porcine aorta. Since, as shown above, the modulus of the elasticity of BVP with an absorbed dose of 0.3 MGy in the “wet state” was the smallest of those studied and this was combined with the lowest compliance, this particular sample was chosen for in vivo testing. Three BVP samples were made for exploration on experimental animals.

### 3.3. In Vivo Study of the of Vascular Prostheses Compliance

The compliance of samples were evaluated in comparison with the compliance of the section of the aorta located proximally from the implantation zone. The results are presented in Table 2.

Analysis of aortograms reveals that the pulsation of the prosthesis remained throughout the entire observation period. Some decrease in the compliance in the middle of the prosthesis was almost no different from a decrease in the compliance of the aorta near the implantation. On the angiograms (Figure 5), it can be seen that the prosthesis retained its original shape, without forming an expansion or stenosis. This indicates the satisfactory thrombogenicity of the material and the mechanical stability of the prosthesis wall under the constant pulse load conditions.

### 3.4. Large-Scale Prosthesis Observation after Extraction, and Histological Study of the Capsule

The condition of the prosthesis capsule and neighboring aorta segments were evaluated visually. On the large-scale specimen, the prosthesis was clearly visualized at the implantation site, was surrounded by a relatively thin capsule, the material of the prosthesis was covered with tightly soldered pseudointima, the inner lumen of the prosthesis was free from blood clots, and the surface was smooth and homogeneous. The inner surface of the prosthesis was uniform in color and thickness and practically did not differ from the inner surface of the biological aorta segments (Figure 6).

A histological examination was carried out on the sections made athwart and along the aorta lumen to evaluate the extent of orientation of the cellular elements of the capsule, which may indicate the possibility of its differentiation under a functional load. The cuts were stained to standard with hematoxylin–eosin.

With standard staining, a layered structure of the internal capsule of the prosthesis was well visible, and the cells of the fibroblastic type in layers were oriented along or across the axis of the prosthesis (Figure 7).

To identify smooth muscle cells in the prosthesis capsule, an immune-chemical staining was carried out. Paraffin sections were incubated with mouse antibodies to the smooth-muscle alpha acting of the pig (1:250, Cell Marque 1a4, Sellmark Corporation, Mansfield, CA, USA) and developed using the immunoperoxidase system (Dako Envision-System-HRP, K4007, Dako Inc., Glostrup, Denmark). The cuts were additionally stained with Mayer hematoxylin.

An immune-chemical study showed a significant amount of smooth muscle cells in the capsule, also oriented in accordance with the directions of deformation of the vessel during circumferential pulse loading (Figure 8). The number of smooth muscle cells in the pseudointima was noticeably greater than in the neoadventitia. There were areas of colonization by the smooth muscle cells of the stroma of the prosthesis (Figure 9).

## 4. Discussion

Modern synthetic aortic prostheses have a sufficiently high thrombogenicity and their mechanical properties differ from that of the blood vessels because their structure cannot imitate the complex structure of the arterial wall [17,18]. The arterial wall consists of three layers: the inner shell, which includes the endothelium and basement membrane; the middle shell is represented by elastic collagen fibers located along the circumference and smooth muscle cells as well as the outer layer containing the connective tissue of collagen fibers oriented longitudinally in the form of a bundle. The mechanical behavior of the aorta is mainly determined by the ratio of the collagen and elastin fibers, which differ significantly in their elasticity (i.e., 0.6–1 MPa for elastin and 1 GPa for collagen). High compliance of the aorta provides a pulsed, swirling mode of the blood flow, which helps to reduce the physiological load on the heart muscle and enables the functioning of the circulatory system [19].

The wall of a synthetic implant should not only mimic the mechanical behavior of the aorta, but should also be porous for successful cell infiltration, biocompatible and antithrombogenic. For the manufacture of aortic prostheses, polyethylene terephthalate, polyesters, polyurethanes, polytetrafluoroethylene, and silk fibroin are generally used [20,21,22,23]. Polytetrafluoroethylene copolymers have attracted great attention due to their suitable mechanical properties for surgical implantation and the possibility of modifying their surface with albumin, collagen, and other biologically active compounds. However, they do not completely reproduce the mechanical properties of a biological arterial vessel. Similarly, low elasticity/compliance of the tissue aortic prostheses made from Dacron^TM^ (1.9 mm Hg × 10^−2^) has a negative impact on their long-term patency [17]. The development of non-woven manufacturing technologies 9e.g., electrospinning or blow molding) has significantly improved the structure (porosity, specific surface area) and mechanical properties (compliance) of aortic prostheses. For example, a polycaprolactone aortic prosthesis with a pore size of up to 41 μm fabricated by electrospinning was implanted in the infrarenal part of the abdominal aorta of a rat [24]. After 12 months, the endothelial and muscle structures of the regenerated prosthesis were similar to that of the native vessel. Regeneration of the vascular bed and restoration of its structure was also reported for tubular prostheses made from silk fibroin by electrospinning [22]. However, in these studies, the mechanical properties of the vessel recovered slowly, which negatively affected the blood flow in the prosthesis area. An attempt to mimic the mechanical properties of native vessels was reported in [25]. For this purpose, tubular polylactide and polyurethane prostheses with adjustable fiber placement were fabricated by electrospinning. Unfortunately, the nonlinear mechanical behavior of the native rat aorta could not be mimicked with any preferred fiber orientation, regardless of the polymer used. Widely used for the manufacture of synthetic vascular prostheses, materials made of polytetrafluoroethylene, polyethylene terephthalate, and polyurethane are rigid, have low biocompatibility, and the stroma of these prostheses is not capable of colonization by cells, which often leads to thrombosis and intimal hyperplasia [26,27]. Modern scientific research is aimed at creating vascular prostheses with certain biomechanical parameters. Therefore, in some studies, the necessary compliance is created with the help of elastin-like recombinamers [28]. The mechanical strength of these prostheses is provided by combining the components of technical textiles made of polyvinylidene fluoride and polycaprolactone [29,30]. According to experiments in “in vitro” conditions, elastin-like vascular prostheses with a porous structure have the necessary strength, and at the same time, their stroma is populated by cells and extracellular matrix [31]. Another group of authors investigated an elastomeric nanofibrillar prosthesis in an experiment in “in vivo” conditions. Compared with a standard polytetrafluoroethylene prosthesis, the synthetic material under study had better endothelization and colonization of stroma cells. Additionally, an elastomeric nanofibrillar prosthesis demonstrated rapid integration into the vascular bed in an experiment on rabbits [32].

However, to date, the problem of achieving the necessary level of long-term mechanical properties of aorta prothesis is still very acute. In present research mechanical properties of the fabricated vessel matched well with that of the native aorta. Moreover, complete epithelialization of the vessel as well as restoration of the muscle layer occurred. In addition, signs of functional differentiation of the cells in the prosthesis capsule (i.e., the appearance of a large number of smooth muscle cells arranged in layers and oriented along the direction of the applied loads (circularly and along the axis of the vessel)) were observed. This functional differentiation creates favorable conditions for restoration of the neovessel tone that is formed in the prosthesis site.

## 5. Conclusions

The influence of radiation treatment on the elastic properties and mechanical stability of fibrous prototypes of BVP obtained by electrospinning was studied. In the course of bench studies, it was shown that at an absorbed dose of 0.3 MGy, the prostheses acquired mechanical optimal properties close to those of the native aorta: their compliance decreased (mechanical stability increased) against the background of a slight decrease in elasticity. The optimal irradiated samples were selected for in vivo experiments. The implantation of BVP with appropriate radial elasticity in the infrarenal segment of the abdominal aorta showed that, in contrast to the previously studied unmodified synthetic prostheses, the connecting capsule formed around the prosthesis had signs of a higher tissue organization. In it, the cells of the fibroblastic series were arranged in layers oriented along and across the prosthesis, similar to the orientation of cells in a biological arterial vessel. Also, in the course of the chronic experiment, it was shown that the radiation-modified samples (in contrast to the original ones studied earlier) retained their original shape, that is, they were mechanically stable.

## Figures and Tables

**Figure 1 jfb-13-00192-f001:**
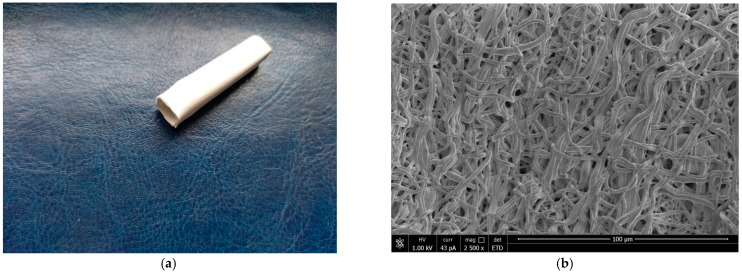
Typical tubular prostheses fabricated by electrospinning (**a**). SEM micrographs of electrospun copolymers of VDF/HFP fibers (**b**).

**Figure 2 jfb-13-00192-f002:**
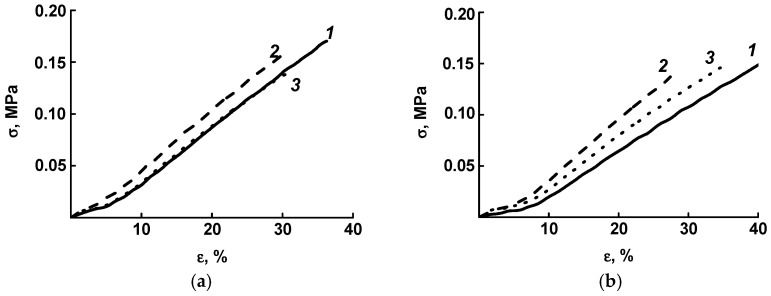
The initial sections of the deformation curves of the “dry” (**a**) and “wet” (**b**) samples of prostheses of the original (1) and radiation-modified 0.1 MGy (2) and 0.3 MGy (3).

**Figure 3 jfb-13-00192-f003:**
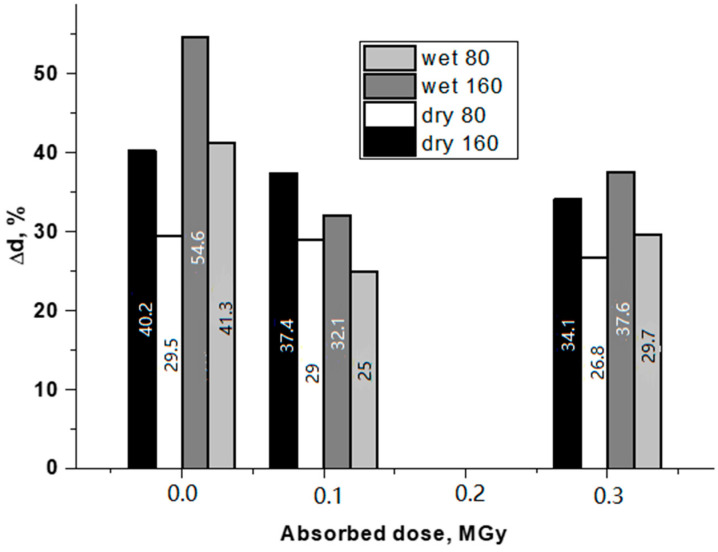
Effect of the absorbed dose on the change in diameter at the maximum (160 mm Hg) and minimum (80 mm Hg) pressures for the “dry” and “wet” prosthesis samples after 10,000 load–unload cycles.

**Figure 4 jfb-13-00192-f004:**
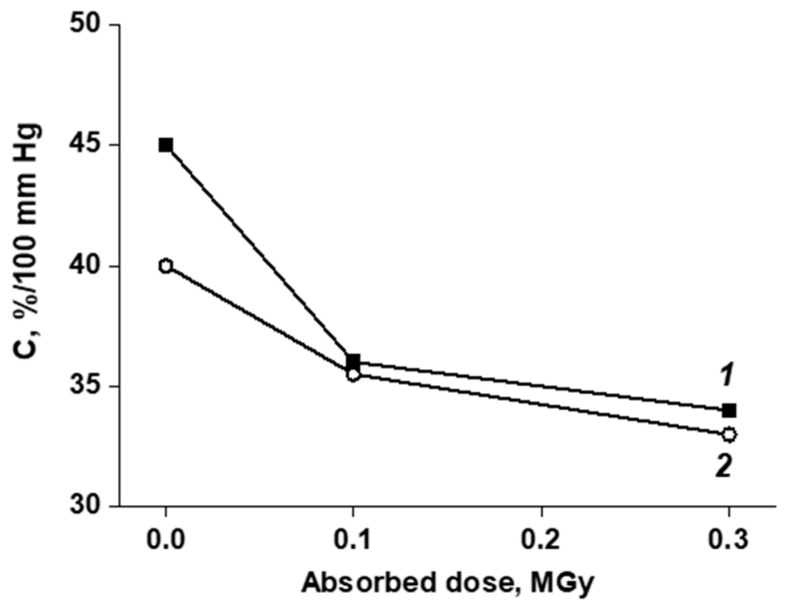
Influence of the absorbed dose on the change in compliance of prototype prostheses after 10,000 load–unload cycles in the pressure range of 80–160 mm Hg. for the “dry” (1) and “wet” (2) samples.

**Figure 5 jfb-13-00192-f005:**
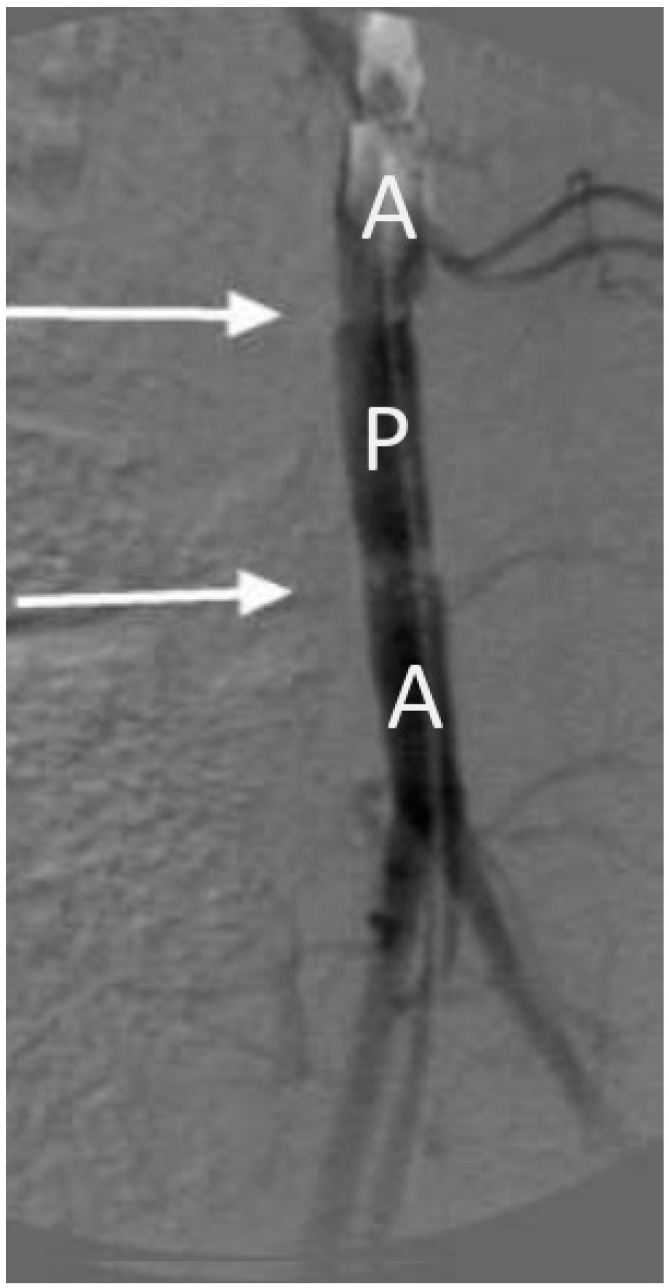
Aortogram of the abdominal aorta with an implanted vessel prosthesis (VDF/HFP (3:1)/VDF/HFP (8:1): 85/15; 0.3 MGy) 1 month after surgery (arrows indicate the sites of anastomosis). A, aorta; A, prosthesis.

**Figure 6 jfb-13-00192-f006:**
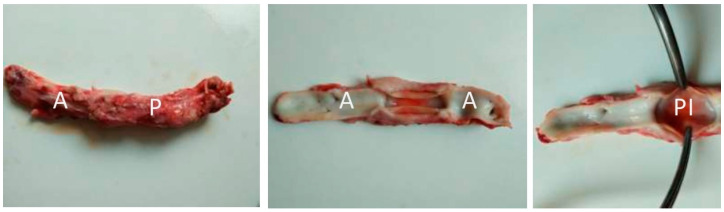
Macropreparations of vascular prosthesis samples in the infrarenal abdominal aorta 1 month after implantation (VDF/HFP (3:1)/VDF/HFP (8:1): 85/15; 0.3 MGy). A, aorta; A, prosthesis; PI, pseudointima.

**Figure 7 jfb-13-00192-f007:**
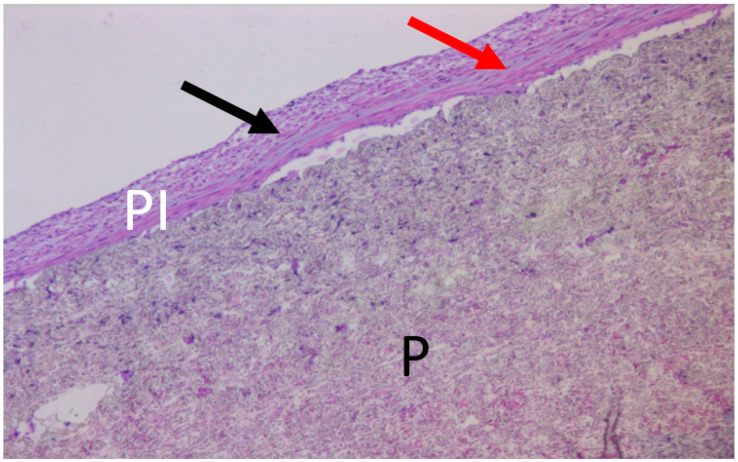
Micrograph of the (VDF/HFP (3:1)/VDF/HFP (8:1): 85/15; 0.3 MGy) prosthetic sample. Staining with hematoxylin and eosin. Magnification, ×100. Smooth muscle cells are located in the longitudinal (black arrow) and transverse directions in relation to the axis of the vessel (red arrow). P, prosthesis; PI, pseudointima.

**Figure 8 jfb-13-00192-f008:**
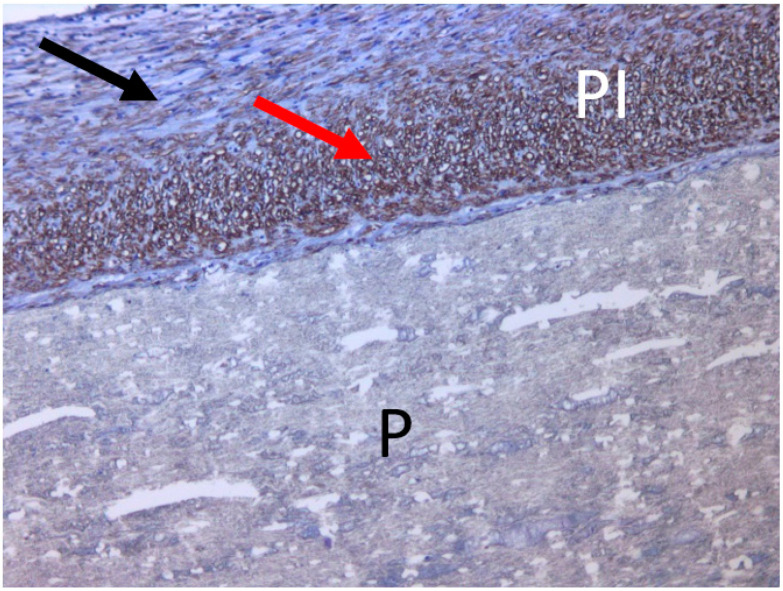
Micrograph of the (VDF/HFP (3:1)/VDF/HFP (8:1): 85/15; 0.3 MGy) prosthetic sample. Immunohistochemical studies. Magnification, ×100. Smooth muscle cells (red arrow) in the inner shell of the prosthesis are located transversely (red arrow) and longitudinally (black arrow). P, prosthesis; PI, pseudointima.

**Figure 9 jfb-13-00192-f009:**
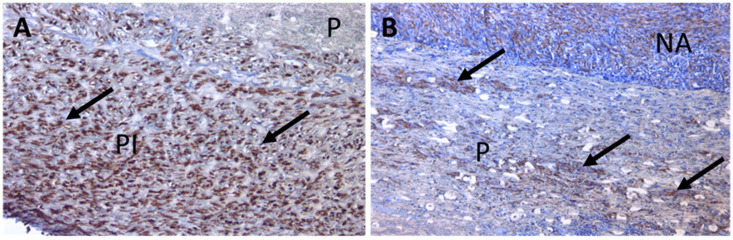
Micrograph of the (VDF/HFP (3:1)/VDF/HFP (8:1): 85/15; 0.3 MGy) prosthetic sample. Immunohistochemical studies. Magnification, ×200. (**A**) Smooth muscle cells (brown) located in the inner shell of the prosthesis (black arrows). (**B**) Smooth muscle cells (brown) located in the stroma of the prosthesis (black arrows) P, prosthesis; PI, pseudointima; NA, neoadventitia.

**Table 1 jfb-13-00192-t001:** Mechanical properties of BVP prototypes based on a mixture of VDF/HFP (3:1) and VDF/HFP (8:1) copolymers.

Sample of BVP	Strength, MPa		Breaking Strain, %		Modulus of Elasticity (10–25%), MPa	
	Dry	Wet	Dry	Wet	Dry	Wet
Control	1.59 ± 0.15	1.32 ± 0.16	722 ± 5	749 ± 15	0.55 ± 0.02	0.45 ± 0.03
0.1 MGy	1.56 ± 0.14	1.54 ± 0.15	653 ± 6	592 ± 7	0.59 ± 0.02	0.60 ± 0.02
0.2 MGy	1.48 ± 0.14	-	551 ± 12	-	0.57 ± 0.02	-
0.3 MGy	1.01 ± 0.11	0.96 ± 0.11	340 ± 5	335 ± 7	0.54 ± 0.02	0.53 ± 0.02
0.4 MGy	0.78 ± 0.09	-	187 ± 7	-	1.42 ± 0.02	-

**Table 2 jfb-13-00192-t002:** Compliance according to the results of angiographic studies.

Sample	Compliance, %/100 mmHg	
	3 Days	30 Days
Native aorta	29 ± 1	24 ± 1
BVP	28 ± 1	22 ± 1

## Data Availability

The data presented in this study are available in the manuscript itself.

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
