# Peer review of "Influence of γ-Radiation on Mechanical Stability to Cyclic Loads Tubular Elastic Matrix of the Aorta"

_jfb, 2022, doi:10.3390/jfb13040192_

Round 1

Reviewer 1 Report

This manuscript addresses a very innovative concept of materials ,a copolymer made of vinylidene fluoride and hexafluoroethylene VDF/HFE cross-linked with gamma radiation.The blood conduits are fabricated by electrospinning.Such a manuscript opens new avenues in cardiovascular biomaterials.Unfortunately,as presented ,it is not attractive as the results are not presented in a very a convincing way.                                                                   1) A description of the vascular grafts shall be given with the appropriate illustrations.2) The results of the in vitro tests shall be detailed. 3) The number of pigs for in vivo investigations is low,only 3.Is that sufficient? Were the grafts patent at the sacrifice of the animals? Some photos of the explanted devices would be appreciated.4) The histology and immunohistology results are poorly described and are not acceptable in 2022.Please refer to the publications of K Berger and LR Sauvage.An illustration of the external capsulae is essential to corroborate the results reported for the compliance .5) The bibliography is not sufficient and the authors must refer to HJ Salacinski,MR Kapadia,F Koffhi,J Joseph,R Guidoin,to name a few.                                                                                                                              Based uopn the innovative characteristics of this blood conduit,this manuscript deserves publication after a major revision is completed.

Reviewer 2 Report

The manuscript describes the influence of g-radiation on mechanical properties of VDH/HFP copolymers for vascular prostheses. Experimental part is well written as the entire paper. Results are well discussed. Reference list is very poor. Authors should be added references in the introduction and results and discussion parts. The state of art and a comparison with materials used for vascular prostheses must be added in the introduction and results and discussion parts, respectively. In my opinion, the paper after minor revision is ready for publication for Journal of Functional Biomaterials.

Reviewer 3 Report

There are some issues of the presentation which should be addressed.  The title mentions fracture resistance as a research object, while the remaining part of manuscript has little presentation of the fracture characteristics.  In introduction section, the original aspects of this study in comparison to the previous ones are not strongly demonstrated.  In addition, the range and number of absorbed doses in the investigation is not large enough for arriving at the conclusion “An optimal absorbed dose of 0.3 MGy was determined.”.  In Figs. 4 and 5, the arrows are not meaningful.  More aortograms and micrographs for comparing the configurations between the original and radiation-modified samples should be placed in the manuscript.

Round 2

Reviewer 1 Report

Accept as it is.This reviewer recommands a more in depth analysis of the histology in subsequant manuscripts.

Reviewer 3 Report

The revised manuscript is greatly improved.  The research is worthy of publication.